# Factors associated with Turkish individuals' purchase of transportation, accommodation, and household services from digital platforms

Hatice Nur Yıldız[1¤a☯], İbrahim Yıldız[2¤b☯], Ömer Alkan [3,4¤c*]

1 Department of Public Relations and Publicity, Ataturk University, Erzurum, Türkiye, 2 Department of Management Information Systems, Ataturk University, Erzurum, Türkiye, 3 Department of Econometrics, Ataturk University, Erzurum, Türkiye, 4 Ata Teknokent, Master Araştırma Eğitim ve Danışmanlık Hizmetleri Ltd. Şti., Erzurum, Türkiye

☯ These authors contributed equally to this work.
¤a Current address: Department of Public Relations and Publicity, Faculty of Communication, Ataturk University, Erzurum, Türkiye
¤b Current address: Department of Management Information Systems, Faculty of Economics and Administrative Sciences, Ataturk University, Erzurum, Türkiye
¤c Current address: Department of Econometrics, Faculty of Economics and Administrative Sciences, Ataturk University, 2nd Floor, Number: 222, Erzurum, Türkiye
* oalkan@atauni.edu.tr

## Abstract

Digital platforms are technological applications with steadily increasing popularity. This study aims to investigate various factors associated with the purchase of transportation, accommodation, and household services via digital platforms by individuals in Türkiye. This study used the 2023 "Household Information Technologies Usage Survey Microdata Set" provided by the Turkish Statistical Institute. To identify the possible effects of factors such as gender, education level, age, income level, and household size on digital platforms, binary logistic regression analysis was used. According to the study's results, all the factors, including gender, education level, age, income level, and household size, may affect the use of digital platforms. Accordingly, the likelihood of using digital platforms is higher among men compared to women, younger individuals compared to older ones, and individuals with higher education and income compared to those with lower levels. In terms of household size, a negative relationship was found. In other words, individuals from larger households are less likely to use digital platforms. The study's findings suggest that beliefs and values, consumption habits, social roles and responsibilities, and the factor of trust in digital platforms play a role in the use of digital platforms. In the study, which includes evaluations that digital platforms will become increasingly widespread in Türkiye, where Western values and a modern lifestyle are gradually becoming more prevalent, recommendations are made regarding the perception and promotion of digital platforms and the importance of public relations practices.

**Data availability statement:** The data underlying this study is subject to third-party restrictions by the Turkish Statistical Institute. Data are available from the Turkish Statistical Institute (bilgi@tuik.gov.tr) for researchers who meet the criteria for access to confidential data. The authors of the study did not receive any special privileges in accessing the data.

**Funding:** The author(s) received no specific funding for this work.

**Competing interests:** The authors have declared that no competing interests exist.

## Introduction

Digital platforms enable transferring products, services, information, or financial value by facilitating interaction between two or more users through robust network infrastructures [1]. According to another definition, digital platforms are complex electronic marketplaces facilitating value-creating interactions between producers and consumers across various industries and geographies [2]. Additionally, digital platforms provide access to products and services without being restricted to a specific geographic region [3]. Furthermore, digital platforms allow individuals to exist in the free market without needing corporate-type structures, enabling them to compete with firms in the same segment by leveraging their skills and assets.

Digital platforms are tools whose use has been steadily increasing alongside developments in information technology infrastructure, enabling economic activities to be carried out online. Through these platforms, parties on the supply and demand sides come together, and goods and services are exchanged digitally [4]. Well-known digital platforms include Airbnb, TaskRabbit, eBay, Netflix, BlaBlaCar, Taobao, and Uber. Each platform can bring together buyers and sellers for different purposes or act as an intermediary in this process. For instance, Airbnb makes unused accommodation spaces for service, while eBay facilitates presenting new and second-hand products. Uber (taxi service), BlaBlaCar (ride sharing), and TaskRabbit (various handyman services) mediate the provision of services for various tasks. Netflix has significantly impacted the film industry, while Taobao is among the platforms used for shopping [5–9].

There are also platforms offering services such as ride-hailing (e.g., DiDi), car sharing (e.g., Zipcar), bike sharing (e.g., Mobike), and fashion sharing (e.g., My Wardrobe HQ) [10]. In addition to these, market leaders such as Amazon, Lyft, and Zillow, as well as Airbnb, Klöckner, and ZBJ, operate across a wide range of sectors, from retail to healthcare services, from real estate to banking, from the steel industry to the labour market [11]. Furthermore, platforms are also being used in the tourism and education sectors (MADDI) [12,13]. Well-known social media platforms such as YouTube and other social networking sites may differ from these platforms, as the distinction between buyer and seller is not as clear-cut. Producers can become consumers at any time on these platforms and vice versa [14].

Digital platforms are not merely systems that facilitate the meeting of buyers and sellers. These platforms are interactive structures that offer personalized experiences deemed valuable by the market and considered worth paying for by buyers, enable consumption without intermediaries, and are driven by interaction [1,15]. Platforms such as Uber, Doordash, Instacart, Ola, Makemytrip, Oyo, and Airbnb are examples of such digital platforms, and digital platforms are playing an increasingly significant role in the global economy [16,17]. For instance, it has been demonstrated that digital platforms are being adopted at an increasing rate in the United States. Various reports indicate that 36% of the U.S. workforce is involved in the freelance economy, and it is predicted that by 2027, more than 50% of the U.S. workforce will participate in economic activities carried out through these platforms [18]. This is because using digital platforms also means that socioeconomically disadvantaged groups

can access entrepreneurial opportunities [19]. In this respect, these platforms have also pioneered certain economic approaches. With the development of digital platforms, approaches such as the sharing economy, gig economy, collaborative economy, collaborative consumption, peer-to-peer economy (P2P), platform economy, access-based economy, and platform-based sharing have also gained recognition in the literature [10].

The gig economy refers to a heterogeneous structure in which various tasks are assigned to workers [20]. The collaborative economy represents an economic perspective in which individuals and institutions, traditionally positioned as consumers or clients in the classical economic system, are now seen as suppliers and service providers. Moreover, the medium of exchange in transactions has diversified, with activities such as the resale, exchange, and even donation of goods, in addition to financial values, coming into play [21]. In the peer-to-peer (P2P) economic model, social dynamics such as co-development and co-prosperity are emphasized. Thus, new economic approaches, cultural perspectives and various regional dynamics are also brought into focus [22]. Peer-to-peer (P2P) services are a subset of sharing economy activities in which consumers can temporarily use privately owned assets (e.g., homes, cars, etc.). In the P2P economic model, in addition to monetary transactions, practices such as donations may also occur [23]. Curtis [24] suggested that these and similar concepts would be more appropriately categorized under the general heading of the sharing economy. In this context, as an inclusive concept, the sharing economy refers to economies built on digital infrastructure where monetary and non-monetary values are exchanged, and social interaction plays a significant role. Indeed, today, the sharing economy is commonly defined as sharing goods and services via Internet platforms [25].

This study investigates the socio-demographic and economic characteristics (age, gender, education level, income level, and household size) associated with purchasing services through digital platforms in Türkiye. In other words, this study was conducted to determine the potential impact of these factors on household services, accommodation, and transportation preferences.

To provide a clearer overview of the study, the methodological framework and significance of the research are briefly summarized here. This study utilizes the 2023 Household Information Technologies Usage Survey Microdata Set obtained from TurkStat, and applies weighted binary logistic regression models to examine how socio-demographic and economic factors—specifically gender, age, education level, income level, and household size—shape individuals' likelihood of purchasing household, transportation, and accommodation services through digital platforms in Türkiye.

The significance of this research lies in its contribution to the limited body of empirical evidence on digital platform usage in middle-income countries. While most existing studies focus on technologically advanced nations, this study provides nationally representative findings for Türkiye and offers interdisciplinary insights into how technological adoption, social norms, digital literacy, and consumption patterns influence engagement with platform-based services. By identifying the groups more or less likely to use digital platforms, the study also provides valuable guidance for policymakers, platform designers, and practitioners seeking to promote equitable and effective digital service adoption.

The primary motivation for conducting this study is to understand the demographic factors—gender, age, education, income level, and household size—that influence the use of digital platforms in Türkiye. Because understanding the factors that influence the use of these digital platforms will enable context-specific evaluations for Türkiye based on parameters such as social norms, levels of digital literacy, trust in digital platforms, and gender roles [26,27]. Indeed, through this study, it will be possible to conduct interdisciplinary assessments of issues closely linked to the level of societal development, such as users' trust in technological platforms, the adoption of data-driven economic approaches, and users' environmental sensitivities [10,23,28,29]. Thus, evaluations of how different social groups approach digital platforms can be further diversified [30]. These evaluations will also encompass assessments based on factors such as shopping and consumption habits and changes in work practices in Türkiye [31]. Therefore, the desire to examine the factors that shape the Turkish society's perspectives toward these systems has constituted the primary motivational basis of this study.

In addition, contributing to the existing literature is among the core motivations of this research. Conducting studies on digital platforms in middle-income countries such as Türkiye holds significant value. Because this is studies on this

topic have predominantly been conducted in technologically advanced countries such as the United States and China. Therefore, focusing specifically on Türkiye in this context is of particular importance [32]. Moreover, despite their contemporary relevance, digital platforms and the economic structures in which they play a central role are widely regarded as an area that has not yet been sufficiently explored [33,34]. The main motivations that led to this research also reveal its importance. Indeed, Ke [35] noted that the socio-economic evaluation of these platforms remains underexplored in the literature. Furthermore, it has been noted that studies conducted on this topic are insufficient both in terms of quantity and content [19]. Thus, since academic studies addressing the economic approaches represented by digital platforms continue to hold significance, this study should be regarded as valuable and important [36,37].

## Literature review

Digital platforms manifest an economic understanding built on interaction and sharing. These platforms also facilitate collaboration among different actors and allow individuals at all levels and businesses of all sizes to participate in markets [38,39]. However, participation in these platforms is affected by certain factors. Being active and competitive on these platforms primarily requires information technology (IT) usage skills. Since digital platforms are based on contemporary technologies such as artificial intelligence, machine learning, and the Internet of Things and prioritize data processing, they have reached dimensions that demand basic IT skills and up-to-date IT competencies. It has been asserted that financial values, data and information function as mediums of exchange on these platforms [20,21,40].

However, the key competencies on these platforms are not limited to IT usage and IT literacy; other factors are also at play. The use of digital platforms is affected by many demographic characteristics. These are gender, age, education and income levels, and household size [41–43]. Regional dynamics, economic indicators, legal and regulatory frameworks, cultural norms and values, and users' perspectives on these elements—each of which interacts with demographic factors—are also among the dynamics that affect the use of digital platforms [10,41–45]. It should also be noted that affinity for current technologies, data literacy, and the ability to interpret data determine individuals' tendencies to engage with digital platforms [22,43,46,47]. Indeed, various studies and reports have demonstrated that individuals with higher levels of education and those living in urban areas are more inclined to use these and similar platforms [48,49].

Hypothesis 1: There is a statistically significant relationship between the level of education and the use of digital platforms.

Horodnic, Williams [49] stated that individuals aged 25–39 who embrace urban life are more willing to use digital platforms. Zhang [50] and Law and Ng [51] highlighted a different aspect by highlighting young individuals' positive and low-anxiety attitudes toward using e-commerce platforms. Similarly, Muda, Mohd [52] emphasized that young individuals are more willing to use these platforms due to their trust in digital platforms.

Hypothesis 2: There is a statistically significant relationship between age and the use of digital platforms.

According to the EU [53] report, men have been shown to use digital platforms more than women. It is possible to argue that gender roles and social norms may play a determining role in the use of digital platforms for both men and women [54,55]. Therefore, changes in gender-based perceptions may also serve as a determining factor in the use of digital platforms. For instance, Ert, Fleischer [56] have provided examples indicating that, in recent periods, women have used digital platforms more than men. Castillo-Villar, Castillo-Villar [57] emphasized the importance of being aware of women's presence on digital platforms and the need to consider their needs.

Hypothesis 3: There is a statistically significant relationship between gender and the use of digital platforms.

Ünver, Aydemir [46] also identified income level as one of the important factors in the use of digital platforms. Accordingly, it can be stated that higher income levels motivate individuals to use digital platforms more. On the other hand, it has been noted that digital platforms, where gig economies are prevalent, often involve irregular income, limited earnings, and insecure working conditions [58]. Indeed, Nathan, Victor [59] indicated that perceived benefits play a significant role in

the use of digital platforms. This situation indicates that individuals can obtain income through a utilitarian approach and, with this determination, they can turn to digital platforms.

Hypothesis 4: There is a statistically significant relationship between the income level factor and the use of digital platforms.

The household factor, influenced by various other factors, has also been found to be significant in the use of digital platforms. For example, [60] revealed that individuals with larger households and higher income levels adopted online shopping during the COVID-19 pandemic. Enam, Azad [61] found that even though families with children use services offered on digital platforms for online shopping, they are 76% more likely to continue physical shopping at the same frequency.

Hypothesis 5: There is a statistically significant relationship between the household size factor and the use of digital platforms.

The influence of these different categories of factors on the use of digital platforms stems from the fact that digital platforms are not merely IT applications. These platforms also have social and psychological dimensions. It is suggested that awareness of fundamental human values such as individual rights, including governance, workers' rights, personal data protection, environmental sensitivity, overconsumption, and poverty may also play a role in shaping attitudes toward these platforms [62–64]. In this regard, Fang and Li [65] included individuals' career decisions and their visions for the future among the factors that may influence their behaviour on these platforms. Pelgander, Öberg [28], by emphasizing the importance of trust in digital platforms, noted that individual competencies may impact trust in digital platforms and thus the likelihood of using them.

This study, which examines the influence of gender, age, education and income level, and household size on the use of digital platforms, is significant in many respects. First, it is noted that the number and content of studies conducted on this subject remain limited [10,19]. Furthermore, this study allows for interdisciplinary evaluations on issues closely related to the level of development, such as the adoption of a data-driven economic perspective, the degree of trust users place in technological platforms, and users' sensitivity to environmental concerns [10,23,28,29]. In addition, since academic studies that address the economic approaches represented by digital platforms continue to be important, this study should be considered both valuable and significant [36,37].

To the best of our knowledge, this study is the first attempt to fill the gap in the literature by exploring how usage behaviours of various digital platforms have developed across Türkiye. This study investigates the various factors associated with individuals' use of digital platforms in Türkiye. In line with this aim, the factors influencing individuals' use of digital platforms have been modelled for Türkiye using a rich data set.

## Method

This section will explain the details of the data used in the study, the dependent and independent variables, and the research method.

### Data

In this study, the "Household Information Technologies Usage Survey Microdata Set," published in 2023 by the Turkish Statistical Institute (TurkStat), was used. The Household Information Technologies Usage Survey, which has been conducted since 2004, aims to collect information about the information and communication technologies possessed and used by households and individuals. In this survey, all settlements across Türkiye were included in the sampling frame. The study covers households located in all residential areas within the borders of Türkiye. However, individuals living in institutional populations such as schools, dormitories, hotels, orphanages, nursing homes, hospitals, and prisons, as well as those residing in military barracks and military guesthouses, were not included. Additionally, settlements for which it was anticipated that a sufficient number of sample households could not be reached, and whose populations would not

exceed 1 percent of the total population (such as small villages, nomadic settlements, and hamlets), were excluded from the scope [66].

The sampling method used in the study was two-stage stratified cluster sampling. In the first stage, clusters (blocks) consisting of an average of 100 households were selected for the sample using the probability proportional to size (PPS) method. In the second stage, sample addresses were determined using the systematic selection method from the selected clusters. The Statistical Regions Classification Level 1 was used as the stratification criterion. The study's sample size was calculated to produce estimates at the Türkiye-total and Statistical Regions Classification Level 1. Due to the consideration of non-response rates in calculating the sample size, no substitution was used in the study. The dataset obtained from the sample was weighted due to selection probabilities in the multi-stage sample design [66].

Data is collected from households selected according to the specified sampling method. The statistical unit used in the Household Information Technology Usage Survey is "households." Demographic information (age, gender) is collected for all members of the household. Questions regarding educational status, labor force status, and information technology usage are asked of members aged 16–74. The Computer-Assisted Personal Interviewing (CAPI) method was used as the data collection method between 2004 and 2019. In 2020 and 2021, Computer-Assisted Telephone Interviewing (CATI) was used, and since 2022, both CAPI and CATI methods have been used together. The survey is conducted every year in April. The reference period is determined according to the week in which the survey is conducted. The 2023 wave of the survey was conducted in April 2023 [66].

The study was carried out on 24,770 individuals aged between 16 and 74 years, and the factors affecting the socio-demographic and economic characteristics (age, gender, education level, income level, and household size) associated with purchasing services through digital platforms in Türkiye were investigated [66].

## Measures

The dependent variables of the study are as follows:

The purchase of household services (such as cleaning, childcare, repair work, gardening, etc.) via websites or mobile applications for private use (Yes, No),

The purchase of transportation services via a website or application for private use through companies (TCDD, THY, Anadolu jet, EGOCepte, Kamil Koç, Pamukkale, ido, budo, bitaksi, etc.) or private individuals (BlaBlaCar, etc.) (Yes, No)

The purchase of accommodation services for private use by booking or renting via a website or application through hotel or tourism agencies (jollytur, etstur, setur, booking, trivago, tatilsepeti, TripAdvisor, etc.) or private individuals (such as Airbnb, Booking) (Yes, No)

The independent variables include gender (male, female), age (16–24, 25–34, 35–44, 45–54, 55–64, and 65+), education level (no formal education/primary school, middle school, high school, university), income level (1st income level being the lowest, 4th income level, 5th income level being the highest), and household size (1–3 persons, 4–5 persons, six persons or more).

## Econometrics method

Survey statistics in Stata 15 (Stata Corporation) were used to account for the complex sampling design and weights. A weighted analysis was performed. First, the frequencies and percentages of the factors associated with purchasing services from different digital platforms by the study participants were obtained. This study used binary logistic regression models to examine the factors associated with purchasing services from various digital platforms.

Non-parametric statistics are used for categorical data (nominal, ordinal). Logistic regression, which is a non-parametric statistical method, is used when the dependent variable is categorical with exactly two outcomes [67,68].

In social sciences, especially in socio-economic research, some of the variables examined are measured on a sensitive scale, while others consist of dichotomous data such as positive-negative, successful-unsuccessful, and yes-no [69].

Dichotomous data are the most commonly used form of categorical data [70]. When the dependent variable is dichotomous categorical data, logistic regression analysis is used to examine the cause-and-effect relationship between the dependent variable and the independent variable(s) [71].

Logistic regression is a statistical method that allows for classification in accordance with probability rules by calculating the predicted values of the dependent variable as probabilities [72].

The logistic model was initially developed for use in survival analysis. Here, the dependent variable (Y) takes values of 1 or 0, depending on whether the event of interest occurs. The expected value, E(Y), never falls below 0 or above 1. Therefore, the predicted values of $\hat{y}$ in the logistic model range between 0 and 1 [73,74].

Logistic model is written as,

$$E(Y) = \pi = P(Y = 1) = \frac{e^{\beta_0 + \beta_1 X_i}}{1 + e^{\beta_0 + \beta_1 X_i}} \quad veya \quad \frac{\exp(\beta_0 + \beta_0 X_i)}{1 + \exp(\beta_0 + \beta_0 X_i)} \tag{1}$$

After dividing the numerator and denominator of Equation 1 by $e^{\beta_0 + \beta_1 X_i}$ or $\exp(\beta_0 + \beta_0 X_i)$,

$$E(Y) = \pi = \frac{1}{1 + e^{-(\beta_0 + \beta_1 X_i)}} \quad or \quad \frac{1}{1 + \exp(-\beta_0 - \beta_0 X_i)} \tag{2}$$

In the equation, there is a condition that $Y = \begin{cases} 1, & \text{if event A occurs} \\ 0, & \text{if event B occurs} \end{cases}$ and X values are qualitative or quantitative independent variables.

Ordinal and nominal variables were defined as dummy variables so that the effects of the categories of all variables to be included in the binary logistic regression model could be observed [75].

## Ethics statement

We declare that all ethical guidelines for authors have been followed by all authors. Ethical approval is not required [76].

TurkStat is an institution that compiles, evaluates, and presents statistical information to decision-makers to prepare development plans and programs, make economic decisions, and address all other issues needed. TurkStat carries out internationally comparable statistical production activities according to the standards of organizations such as the European Union Statistical Office, the United Nations, OECD, ILO, etc. TurkStat collects data within the scope of the Official Statistics Program. The Official Statistics Program is prepared for five-year periods based on the Turkish Statistics Law No. 5429 to determine the basic principles and standards regarding the production and publication of official statistics and to ensure the production of up-to-date, reliable, timely, transparent and impartial data in areas of need at national and international levels [76]. The individuals included in the sample in the surveys conducted by the Turkish Statistical Institute are legally obliged to participate in the survey. According to the Turkish Statistics Law, those who do not provide the information requested within the scope of the research in the specified form and time without a valid excuse or who provide incomplete or incorrect information are warned once and asked to provide the information or correct the deficiencies and errors within seven days. An administrative fine shall be imposed on natural persons or organs and representatives of private legal entities who, despite this warning, do not provide the information or correct the deficiencies and errors as requested [76]. TurkStat also conducts the Household Information Technologies Usage Survey within the scope of the Official Statistics Program put into effect by law. Since the Household Information Technologies Usage Survey Microdata Set is conducted within the scope of legal responsibility by the state, ethical approval is not required [77].

The data were obtained through the joint teamwork of both the Turkish Statistical Institute (TurkStat) and the European Union Statistical Office (SOEU). We obtained this data from TurkStat in return for a contract without needing an ethics committee document and used it in our study [76].

## Results

Table 1 presents the findings related to the use of the online service sector. When Table 1 is examined, it is seen that among those who benefited from the digital services covered in the study, 50.6% are women and 49.4% are men. Among those who preferred the digital service sector, 20.8% are in the 35–44 age group, and 38.4% have an education level of primary school or no formal schooling. In addition, 20.6% of individuals who preferred digital services are in the third income level, and 50.6% belong to households with a size of 1–3 persons.

The multicollinearity among the independent variables included in the binary logistic regression model was tested [78]. It is stated that variables with a Variance Inflation Factor (VIF) of 5 or above indicate moderate multicollinearity [79]. In contrast, those with a VIF of 10 or above indicate a high degree of multicollinearity [80,81]. In this study, no independent variable was found to cause a multicollinearity problem.

The results of the estimated binary logistic regression model are presented in Table 2.

When Table 2 is examined, it is observed that age, education level, and income level are significant variables affecting household services (such as cleaning, childcare, repair work, gardening, etc.) obtained via websites or mobile applications for private use.

When examining whether demographic factors affect the use of institutional transportation services (TCDD, THY, Anadolu jet, EGOCepte, Kamil Koç, Pamukkale, ido, budo, bitaksi, etc.) or private individuals (BlaBlaCar, etc.) via a website or application for private use, it is understood that gender, age, education level, income level, and household size are significant variables in e-transportation service preferences.

In terms of e-accommodation, gender, age, education level, income level, and household size are also found to be significant variables influencing the use of accommodation services either through booking or renting via a website or

**Table 1. Findings regarding factors related to individuals purchasing services from digital platforms.**

| Variables | | Frequency | Percentage | VIF |
|---|---|---|---|---|
| **Gender** | Male | 12,244 | 49.4 | 1.03 |
| | Female | 12,526 | 50.6 | Ref. |
| **Age** | 16-24 | 4,009 | 16.2 | 2.82 |
| | 25-34 | 4,798 | 19.4 | 2.84 |
| | 35-44 | 5,142 | 20.8 | 2.79 |
| | 45-54 | 4,459 | 18.0 | 2.38 |
| | 55-64 | 3,739 | 15.1 | 2.08 |
| | 65 and above | 2,623 | 10.6 | Ref. |
| **Education** | No formal education/primary school | 9,524 | 38.4 | Ref. |
| | Secondary school | 4,074 | 16.4 | 1.52 |
| | High school | 5,915 | 23.9 | 1.66 |
| | University | 5,257 | 21.2 | 1.82 |
| **Income Level** | 1st income level (lowest) | 4,740 | 19.1 | Ref. |
| | 2nd income level | 4,944 | 20.0 | 1.72 |
| | 3rd income level | 5,096 | 20.6 | 1.77 |
| | 4th income level | 4,921 | 19.9 | 1.81 |
| | 5th income level | 5,069 | 20.5 | 1.99 |
| **Household Size** | 1-3 persons | 12,538 | 50.6 | 2.87 |
| | 4-5 persons | 9,213 | 37.2 | 2.61 |
| | 6 persons and above | 3,019 | 12.2 | Ref. |

**Table 2. Estimated model results of the factors associated with the use of digital platforms.**

| Variables | Purchasing household services from digital platforms | | Purchasing transportation services from digital platforms | | Purchasing accommodation services from digital platforms | |
|---|---|---|---|---|---|---|
| | β | Std. Error | β | Std. Error | β | Std. Error |
| **Constant** | −8.335[a] | 0.805 | −6.845[a] | 0.291 | −8.589[a] | 0.518 |
| **Gender (references: female)** | | | | | | |
| Male | 0.22 | 0.115 | 0.200[a] | 0.056 | 0.316[a] | 0.091 |
| **Age (references: 65 and above)** | | | | | | |
| 16-24 | 1.649[b] | 0.772 | 2.213[a] | 0.236 | 1.395[a] | 0.412 |
| 25-34 | 2.600[a] | 0.752 | 1.932[a] | 0.231 | 2.022[a] | 0.393 |
| 35-44 | 2.537[a] | 0.755 | 1.676[a] | 0.234 | 1.842[a] | 0.396 |
| 45-54 | 2.172[a] | 0.758 | 1.489[a] | 0.237 | 1.529[a] | 0.402 |
| 55-64 | 0.995 | 0.813 | 0.886[a] | 0.256 | 0.797[c] | 0.442 |
| **Education (references: no formal education/primary school)** | | | | | | |
| Secondary school | 0.907[a] | 0.349 | 0.846[a] | 0.170 | 0.430 | 0.342 |
| High school | 1.544[a] | 0.300 | 1.897[a] | 0.147 | 1.795[a] | 0.276 |
| University | 2.187[a] | 0.291 | 2.718[a] | 0.145 | 2.649[a] | 0.269 |
| **Income Level (references: 1st income level (lowest))** | | | | | | |
| 2nd income level | −0.236 | 0.324 | 0.002 | 0.132 | 0.184 | 0.294 |
| 3rd income level | 0.325 | 0.285 | 0.297[b] | 0.124 | 0.330 | 0.281 |
| 4th income level | 0.363 | 0.278 | 0.588[a] | 0.119 | 0.988[a] | 0.266 |
| 5th income level | 0.602[b] | 0.274 | 0.985[a] | 0.115 | 1.656[a] | 0.257 |
| **Household Size (references: 6 persons and above)** | | | | | | |
| 1-3 persons | 0.311 | 0.270 | 0.752[a] | 0.119 | 0.878[a] | 0.245 |
| 4-5 persons | 0.322 | 0.265 | 0.375[a] | 0.117 | 0.419[c] | 0.246 |

[a]p < .01; [b]p < .05; [c]p < .10

application from hotels or tourism agencies (jollytur, etstur, setur, booking, trivago, tatilsepeti, TripAdvisor, etc.) or private individuals (such as Airbnb, Booking) within the last three months for private purposes.

The marginal effects of the factors associated with the preference for online services are presented in Table 3.

When Table 3 is examined, it is seen that individuals in the age groups 16–24, 25–34, 35–44, and 45–54 are more likely to purchase household services via digital platforms compared to those aged 65 and over, by 164.2%, 257.8%, 251.6%, and 215.8%, respectively. The likelihood of purchasing household services through digital platforms is 90.2% higher for middle school graduates, 153.1% higher for high school graduates, and 215.9% higher for university graduates than individuals with no schooling or only a primary school education. Individuals in the highest income group are 59.2% more likely to purchase household services through digital platforms than those in the lowest income group.

Men are 18.1% more likely than women to purchase transportation services via digital platforms. Individuals aged 16–24, 25–34, 35–44, 45–54, and 55–64 are more likely to use digital platforms for transportation services compared to those aged 65 and above, by 207.2%, 182.5%, 159.5%, 142.4%, and 85.8%, respectively. The likelihood of purchasing transportation services through digital platforms is 82.4% higher for middle school graduates, 180.9% higher for high school graduates, and 251.4% higher for university graduates than individuals with no schooling or only a primary school education. Individuals in the 3rd, 4th, and 5th income groups are more likely to purchase transportation services than those in the lowest income group, by 27.7%, 54.2%, and 89.5%, respectively. Individuals from households with 1–3 and

**Table 3. Marginal effects regarding the use of digital platforms.**

| Variables | Purchasing household services from digital platforms | | Purchasing transportation services from digital platforms | | Purchasing accommodation services from digital platforms | |
|---|---|---|---|---|---|---|
| | ME | Std. Error | ME | Std. Error | ME | Std. Error |
| **Gender (references: female)** | | | | | | |
| Male | 0.022 | 0.113 | 0.181[a] | 0.050 | 0.306[a] | 0.088 |
| **Age (references: 65 and above)** | | | | | | |
| 16-24 | 1.642[b] | 0.771 | 2.072[a] | 0.229 | 1.376[a] | 0.41 |
| 25-34 | 2.578[a] | 0.750 | 1.825[a] | 0.225 | 1.983[a] | 0.39 |
| 35-44 | 2.516[a] | 0.754 | 1.595[a] | 0.228 | 1.810[a] | 0.393 |
| 45-54 | 2.158[a] | 0.756 | 1.424[a] | 0.231 | 1.506[a] | 0.398 |
| 55-64 | 0.992 | 0.811 | 0.858[a] | 0.249 | 0.790[c] | 0.438 |
| **Education (references: no formal education/primary school)** | | | | | | |
| Secondary school | 0.902[a] | 0.347 | 0.824[a] | 0.166 | 0.427 | 0.34 |
| High school | 1.531[a] | 0.298 | 1.809[a] | 0.143 | 1.771[a] | 0.274 |
| University | 2.159[a] | 0.289 | 2.514[a] | 0.140 | 2.589[a] | 0.267 |
| **Income Level (references: 1st income level (lowest))** | | | | | | |
| 2nd income level | −0.234 | 0.321 | 0.002 | 0.125 | 0.181 | 0.294 |
| 3rd income level | 0.320 | 0.282 | 0.277[b] | 0.116 | 0.326 | 0.277 |
| 4th income level | 0.358 | 0.275 | 0.542[a] | 0.111 | 0.969[a] | 0.262 |
| 5th income level | 0.592[b] | 0.270 | 0.895[a] | 0.107 | 1.609[a] | 0.253 |
| **Household Size (references: 6 persons and above)** | | | | | | |
| 1-3 persons | 0.306 | 0.267 | 0.687[a] | 0.111 | 0.854[a] | 0.24 |
| 4-5 persons | 0.317 | 0.262 | 0.348[a] | 0.110 | 0.410[c] | 0.242 |

[a] $p < .01$; [b] $p < .05$; [c] $p < .10$

4–5 members are more likely to use digital platforms for transportation services than those with six or more members, by 68.7% and 34.8%, respectively.

Men are 30.6% more likely than women to purchase accommodation services through digital platforms. Individuals aged 16–24, 25–34, 35–44, 45–54, and 55–64 are more likely to purchase accommodation services compared to those aged 65 and over, by 137.6%, 198.3%, 181.0%, 150.6%, and 79.0%, respectively. The likelihood of purchasing accommodation services is 177.1% higher for high school graduates and 258.9% higher for university graduates than individuals with no schooling or only a primary school education. Individuals in the 4th and 5th income groups are more likely to purchase accommodation services through digital platforms than those in the lowest income group, by 96.9% and 160.9%, respectively. Individuals from households with 1–3 and 4–5 members are more likely to purchase accommodation services via digital platforms than those with six or more members, by 85.4% and 41.0%, respectively.

## Discussion

This study investigates the socio-demographic and economic characteristics (age, gender, education level, income level, and household size) associated with purchasing services through digital platforms in Türkiye. In other words, this study was conducted to determine the potential impact of these factors on household services, accommodation, and transportation preferences.

This study is valuable both in terms of its subject matter and the population it examines. This is because the study enables evaluations of various demographic groups in a developing country like Türkiye, both in terms of their attitudes

toward digital technologies and their perspectives on the sharing economy. In addition, the study addresses the research need concerning collaborative consumption practices among middle-class consumers [82].

The study's findings suggest that gender may be a factor in the use of accommodation and transportation services. It was found that men are more likely than women to use these services through digital platforms. The nature and content of services accessed via digital platforms may be shaped by whether the service is perceived as masculine or feminine. Gendered perspectives and women's societal and familial roles may impose limiting effects on their presence and participation in digital platforms [83]. In this context, trust in digital platforms and anxiety levels should also be considered influential factors [28,50–52]. In terms of confidence in these factors, men may act bolder than women, and the structure of Turkish society, including its beliefs, values, and norms, may encourage men to engage in service purchases via digital platforms with greater courage and lower anxiety. Cheng, Mou [84] reported that privacy concerns and attitudes toward confidentiality may serve as determining factors in the use of digital platforms.

Age also emerges as a significant variable in the use of digital platforms. According to the findings, individuals aged 25–34 are most inclined to benefit from services such as transportation and accommodation via digital platforms, while this tendency decreases with age. Individuals aged 55–64 are seen to use digital platforms only for transportation and accommodation services and not for household services. Trust and anxiety, consumption habits, feelings of stress, and digital literacy skills are suggested to play a role in this result, indicating that older individuals may maintain a more distant attitude toward digital platforms [28,50–52,85]. Blut and Wang [86] noted that young individuals' more modern assessments of financial realities, compared to older individuals, may also influence their use of digital platforms. Indeed, these authors have also reported that contemporary trends—such as possessing environmental sensitivities—may likewise influence behaviors on digital platforms.

The study also revealed that education level may be influential in using digital platforms. It was found that the likelihood of using digital platforms also rises as the education level increases. Specifically, only high school and university graduates were found to have a significantly higher and more pronounced likelihood of using digital platforms for accommodation services compared to primary school graduates. However, in the case of household and transportation services, individuals with middle school education were also found to be statistically significant users. It is suggested that individuals with higher education levels, career goals, broader perspectives, and strong vision are more inclined to engage with digital platforms [65]. In addition, the greater openness of educated individuals to innovations may also influence their use of digital platforms [86]. Alkan, Küçükoglu [48] also revealed that individuals with higher education levels are more prominent in using e-commerce applications. As a result, the high level of digital literacy among educated individuals and their detachment from traditional consumption habits due to a modern lifestyle in their professional and personal lives may be decisive factors.

According to the income level findings, individuals with higher income levels are more likely to use digital platforms for household, transportation, and accommodation services. Individuals in the highest income group are significantly more likely to benefit from these services through digital platforms. Research findings suggest that income level can be a determining factor in using digital platforms. Ünver, Aydemir [46] noted the positive effect of higher income levels on shopping through e-commerce platforms. On the other hand, it has been revealed that individuals who offer services on gig platforms mostly consist of those with lower income levels [87]. These results indicate that utilitarian motivations may be prominent in digital platforms. Based on this, it seems reasonable to assume that individuals with higher income levels, who are more active in their professional lives, are also more likely to use digital platforms.

Finally, household size has also been identified as a significant factor in preferences for digital platforms. As household size decreases (in households with 1–3 members), the likelihood of using digital platforms for transportation and accommodation services increases. This suggests that individuals in traditional family structures may be more distant from digital platforms, while those in modern nuclear family models may be more inclined to use them. Enam, Azad [61] found that family members in larger households may be more inclined toward traditional physical shopping. Stranahan and Kosiel

[88], along Eriksson and Stenius [60], also stated that in segments where household size is a prominent factor, consumption habits and trends observed during specific periods may, along with income level and other factors, influence the tendency to use digital platforms. Horodnic, Williams [49] also reported that embracing urban life can affect the shift toward digital platforms. Therefore, it is reasonable to assume that individuals from large households less associated with a modern, urban profile may not yet have fully adopted digital platforms. Indeed, it has been shown that in Türkiye, regions with higher population density, internet access, and users exhibit a stronger tendency to engage with digital platforms [89]. This indicates that urbanization may serve as a determining factor in the use of digital platforms.

Theoretically, this study contributes to the growing literature on digital platform usage by providing nationally representative evidence from a middle-income country context, where empirical studies remain limited. By modelling the effects of socio-demographic and economic variables on the likelihood of using platform-based services, the study extends existing theoretical frameworks on technology acceptance, digital inequality, and the sharing economy. The findings underscore the importance of social norms, digital literacy, economic stratification, and lifestyle differences in shaping technology-mediated consumption behaviours, thereby offering an enriched understanding of the determinants of digital platform adoption in non-Western societies.

Practically, the results provide actionable insights for policymakers, public institutions, and platform operators. Identifying demographic groups that are less likely to use digital platforms—such as older adults, individuals with lower education and income, and those from larger households—can guide targeted interventions to reduce digital disparities. These findings highlight the need for strategies that enhance digital literacy, build trust in platform-based services, and improve outreach to vulnerable groups. Furthermore, understanding the patterns of service use across household, transportation, and accommodation sectors may help businesses design more inclusive user experiences and communication campaigns tailored to Türkiye's socio-cultural context.

This study has several limitations that should be acknowledged. First, the analysis relies on cross-sectional microdata from the 2023 TurkStat survey, which restricts the ability to infer causal relationships. Second, the dataset includes self-reported behavioural measures, which may be subject to recall bias or social desirability bias. Third, the variables included in the survey limit the scope of the analysis, as important psychological constructs—such as trust, perceived risk, digital literacy level, or attitudes toward platforms—could not be directly measured. Additionally, the study focuses solely on Türkiye, which may limit the generalizability of the findings to other cultural or economic contexts.

## Conclusion

The findings of this study indicate that factors related to beliefs and values, consumption habits, the ability to use digital technologies, perceptions of ownership and property, understandings of individual freedom, and the changing profile of the Turkish workforce may affect the use of digital platforms for service procurement. Furthermore, the research, which also shows that digital platforms may become increasingly accepted in parallel with societal changes in Türkiye, offers insights for individuals and companies providing services via digital platforms. Indeed, it can be argued that access to digital platforms alone may not be sufficient to influence their usage [90].

The fact that gender appears as a significant factor in access to accommodation and transportation services via digital platforms and that men are more likely to use these platforms than women may reflect gender-based concerns among Turkish consumers regarding such services. In Türkiye, where conservative values are prominent, gender-related concerns may pose a barrier to women's equal use of these platforms. On the other hand, the statistically insignificant presence of women in household service procurement through digital platforms, contrasted with their relatively higher demand for transportation and accommodation services, may reflect women's increasing participation in the labour force. In this context, the findings that higher income and education levels increase the likelihood of using digital platforms for household, transportation, and accommodation services and that individuals in higher income groups are more likely to adopt

these technologies, while those with higher education levels are more inclined to use digital platforms for accommodation services, suggest that participation in professional life and being positioned in higher social strata positively influence individuals' attitudes toward digital platforms.

The potential of age to influence nearly all variables highlights the role of IT literacy and familiarity with digital technologies in facilitating the use of digital platforms. The fact that younger individuals are more prominent in using digital platforms for accommodation and transportation services may reflect a shift in mindset, where concepts such as personal freedom and flexible approaches to ownership override traditional values. Moreover, the growing isolation among young people in Turkish society may also drive their engagement with digital platforms for household, accommodation, and transportation services. These findings suggest that marital status may also be an influential factor. The observed decrease in the likelihood of using digital platforms for accommodation and transportation as household size increases supports this interpretation.

Consumption habits also play a role in the use of digital platforms. The lack of a statistically significant effect of household size and gender on the use of digital platforms for household services may indicate that traditional modes of operation still prevail over digital alternatives. The traditional belief in larger families that women should perform household services may delay the perceived necessity of procuring these services through digital platforms.

The study's findings suggest that services offered through digital platforms will likely become more widespread in Turkish society. This is because Turkish society is transforming Western norms. This transformation spans a wide spectrum, from changes in family structure to shifts in consumption habits. The socio-cultural factors influencing the use of digital platforms may also be relevant in the Turkish context. Accordingly, these findings point to the importance, in the near future, of implementing perception-building and public communication strategies for individuals and institutions offering services via digital platforms. Increasing education and awareness levels related to IT use, adopting approaches that emphasize governance and transparency in digital service processes, and developing effective public relations programs will likely enhance the effective use of these platforms. Indeed, promoting digital platforms, removing perceptions and attitudes rooted in distrust, and communicating the benefits these platforms can offer will be of critical importance [91–93]. Furthermore, regulating sharing platforms and the services they provide, as well as implementing relevant legal frameworks, will also prove effective [84]. In this context, identifying the factors that influence the success of public relations practices on digital platforms, as well as uncovering the cultural norms, values, and belief systems that shape attitudes toward these platforms, represent valuable research topics for future studies. Future research directions suggested by this study may also include examining the nature of any security concerns related to digital platforms, the adequacy of the legal framework governing their operation, and the level of knowledge and awareness of digital platforms among public institutions and organizations in Türkiye.

## Author contributions

**Conceptualization:** Ömer Alkan.

**Data curation:** Ömer Alkan.

**Formal analysis:** Ömer Alkan.

**Investigation:** Hatice Nur Yıldız.

**Methodology:** Hatice Nur Yıldız.

**Validation:** İbrahim Yıldız.

**Visualization:** İbrahim Yıldız.

**Writing – original draft:** Hatice Nur Yıldız, İbrahim Yıldız.

**Writing – review & editing:** Hatice Nur Yıldız, İbrahim Yıldız, Ömer Alkan.

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
