## [Decision Letter · Decision Letter 0]

28 Oct 2025

Dear Dr. Alkan,

Thank you for submitting your manuscript to PLOS ONE. After careful consideration, we feel that it has merit but does not fully meet PLOS ONE’s publication criteria as it currently stands. Therefore, we invite you to submit a revised version of the manuscript that addresses the points raised during the review process.

We look forward to receiving your revised manuscript.

Kind regards,

Tachia Chin

Academic Editor

PLOS ONE

Journal Requirements:

2. For studies involving third-party data, we encourage authors to share any data specific to their analyses that they can legally distribute. PLOS recognizes, however, that authors may be using third-party data they do not have the rights to share. When third-party data cannot be publicly shared, authors must provide all information necessary for interested researchers to apply to gain access to the data. (https://journals.plos.org/plosone/s/data-availability#loc-acceptable-data-access-restrictions)

**Additional Editor Comments:**

Please revise the manuscript according to the reviewers' comments.

Reviewers' comments:

Reviewer's Responses to Questions

**Comments to the Author**

1. Is the manuscript technically sound, and do the data support the conclusions?

Reviewer #1: Yes

Reviewer #2: Yes

Reviewer #3: Yes

2. Has the statistical analysis been performed appropriately and rigorously?

Reviewer #1: Yes

Reviewer #2: Yes

Reviewer #3: Yes

3. Have the authors made all data underlying the findings in their manuscript fully available?

Reviewer #1: Yes

Reviewer #2: Yes

Reviewer #3: Yes

4. Is the manuscript presented in an intelligible fashion and written in standard English?

Reviewer #1: Yes

Reviewer #2: Yes

Reviewer #3: Yes

Reviewer #1: Thank you for this opportunity to review your Manuscript. Unfortunately, I found crucial weaknesses affecting the paper that caused me to reject it. This article investigated various factors associated with the purchase of transportation, accommodation, and household services via digital platforms by individuals in Türkiye. Although it provides some novel insights, this study still has some shortcomings:

1.Introduction

(1)The introduction section lacks research background or motivation. I suggest briefly explaining the importance of digital platforms or the research background at the beginning.

(2)The current introduction only describes the definition and phenomena of digital platforms, but does not elaborate in detail on the specific impact of the current situation on individuals. This raises questions about the urgency and significance of the research, making it hard to understand why this study is so crucial and which specific gaps it attempts to fill in the current knowledge system. Therefore, I suggest clearly pointing out the important role of digital platforms and clarifying the research gap that this article aims to fill.

(3)The current introduction lacks the section on research methods and significance. It is suggested that the methods and significance of this research be briefly expounded to make the article more complete.

2.Literature Review

Research significance and research gaps will be more reasonable in the introduction. It is suggested to adjust the structure of the article to make it more in line with academic norms.

3.Data

The data description should systematically cover information such as data sources, survey time, sample size, and data collection methods. It is recommended to elaborate on this part specifically, which can significantly enhance the reliability of the research results.

4.Discussion

(1)The depth of the argument is insufficient. It only stays at the level of correlation without exploring the causal relationship and fails to put forward a strong and well-documented argument to explain why these relationships exist. For example, gender: Is it because men are more adventurous? Or is it because the platform's algorithm itself has gender bias? Or are women holding back due to safety concerns? More specific mechanisms are needed rather than general discussions about "social structure". For instance, age: Is it because of low digital literacy, lack of demand, or because the interface design is not friendly to the elderly that the elderly keep away from digital platforms? Therefore, it is suggested that they be distinguished and discussed in depth.

(2)This part lacks theoretical significance, practical significance, research limitations and future research directions.

Finally,The article lacks a theoretical basis and research hypothesis section, has an unreasonable structure arrangement, and weak theoretical support. It is suggested to supplement relevant theories to support the article.I suggest the author to improve their manuscript by reviewing and adding recent literature, and finally, I would like to encourage the author to improve their manuscript. I believe that the above points need improvement I once again congratulate the authors of this study and hope that my comments are constructive and helpful for the development of your manuscript.

Best regards.

Reviewer #2: Thank you for the opportunity to review this manuscript. I outline minor suggestions below:

- There are few studies examining consumer-level factors in digital platform services (transportation, accommodation, home services) in the Turkish context. In this respect, this study contributes to the literature and has high local specificity.

- The findings are valuable for understanding the usage dynamics of digital platform services in Turkey, generating directly applicable insights for policymakers and industry (e.g., the impact of factors such as age, income, and education).

- The application of binary logistic regression is well-documented, yet the manuscript lacks comprehensive detail on the sampling methodology.

- After reviewing the paper, I find that it adeptly tackles the pertinent issues at hand. However, as a scientific document, it is imperative for the methodology to be clearly delineated. Despite providing a descriptive account of the method, I strongly suggest incorporating an exposition of the applied logistic regression, including its mathematical formulation. This enhancement will enhance clarity significantly. Once this aspect is addressed, the paper will notably distinguish itself.

Reviewer #3: This study is a timely and important study that examines the socio-demographic factors influencing service purchasing behavior through digital platforms, specifically in Turkey. The use of the 2023 "Household Information Technology Usage Survey Micro Dataset" provides a strong empirical basis for the study. Furthermore, the binary logistic regression analysis used is an appropriate method for revealing the relationship between the identified variables and platform usage.

Overall, the study offers significant contributions to the literature with its topic, method, dataset, and the timeliness of its findings. It is a valuable resource for both the academic community and public policymakers in understanding and guiding Türkiye's digitalization process.

My recommendations for the study are below.

*It is recommended that you briefly mention the binary logistics method in the Econometrics Method section.

*In the conclusion section, the limitations of the study can be mentioned in the last paragraph.

**Do you want your identity to be public for this peer review?** For information about this choice, including consent withdrawal, please see our Privacy Policy

Reviewer #1: No

Reviewer #2: No

Reviewer #3: **Yes:** Şeyda Ünver

---

## [Author Response · Author response to Decision Letter 1]

10 Dec 2025

Dear Editor and Reviewers,

Thank you very much for your comments concerning our manuscript titled “Factors associated with Turkish individuals’ purchase of transportation, accommodation and household services from digital platforms”. These comments have been very helpful in reviewing and improving our manuscript. We have carefully revising these instructive comments and made corrections that we hope will be approved. The revised parts are highlighted in red in the main paper.

The corrections to the manuscript and responses to the reviewer’s comments are as following:

Regards,

Authors

Reviewer #1:

Comment: Thank you for this opportunity to review your Manuscript. Unfortunately, I found crucial weaknesses affecting the paper that caused me to reject it. This article investigated various factors associated with the purchase of transportation, accommodation, and household services via digital platforms by individuals in Türkiye. Although it provides some novel insights, this study still has some shortcomings:

Response: Thank you for the comments.

Comment 1:

1. Introduction

(1) The introduction section lacks research background or motivation. I suggest briefly explaining the importance of digital platforms or the research background at the beginning.

Response 1: Thank you for the comments. We substantially revised the beginning of the Introduction to provide a clearer research background and motivation. Specifically, we added a new paragraph explaining the growing importance of digital platforms in everyday life, their expanding role in transportation, accommodation, and household services, and the socioeconomic changes associated with their adoption. We also clarified the relevance of examining digital platform usage in Türkiye by emphasizing its rapid digitalization and the increasing penetration of platform-based services.

Comment 2:

The current introduction only describes the definition and phenomena of digital platforms, but does not elaborate in detail on the specific impact of the current situation on individuals. This raises questions about the urgency and significance of the research, making it hard to understand why this study is so crucial and which specific gaps it attempts to fill in the current knowledge system. Therefore, I suggest clearly pointing out the important role of digital platforms and clarifying the research gap that this article aims to fill.

Response 2: Thank you for the comments. In accordance with your comment, we have substantially revised the Introduction to more clearly present the urgency, significance, and individual-level impact of the rise of digital platforms.

Comment 3: The current introduction lacks the section on research methods and significance. It is suggested that the methods and significance of this research be briefly expounded to make the article more complete.

Response 3: Thank you for the comments. In accordance with your suggestion, we have revised the Introduction by adding a concise explanation of the research method and a brief statement highlighting the significance and contribution of the study (see the last paragraph of the Introduction). These additions improve the coherence of the introduction and provide readers with a clearer understanding of the methodological framework and the relevance of our research.

Comment 4:

2. Literature Review

Research significance and research gaps will be more reasonable in the introduction. It is suggested to adjust the structure of the article to make it more in line with academic norms.

Response 4: Thank you for the comments. Following your suggestion, we revised the structure of the manuscript to ensure that the research significance and research gaps are clearly and appropriately presented in the Introduction, in line with academic conventions. In the revised version, the Introduction now includes an explicit explanation of (1) the importance of digital platforms in shaping individual consumption behavior, (2) the urgency and relevance of examining demographic differences in platform use, and (3) the specific research gap addressed by this study.

Accordingly, the Literature Review section was streamlined to focus solely on summarizing prior empirical findings and building the conceptual foundations for the hypotheses, without repeating the research significance or gap.

Comment 5:

3. Data

The data description should systematically cover information such as data sources, survey time, sample size, and data collection methods. It is recommended to elaborate on this part specifically, which can significantly enhance the reliability of the research results.

Response 5: Thank you for the comments. We carefully revised the Data section to provide a more systematic and comprehensive description of the dataset. In the revised manuscript, we now clearly specify:

Data source: The study uses the 2023 Household Information Technologies Usage Survey Microdata Set obtained from TurkStat.

Survey time: We explicitly state that the 2023 wave of the survey was conducted in April 2023.

Sample size: We clarify that the final sample includes 24,770 individuals aged 16–74, representing Türkiye’s household population.

Data collection methods: We elaborate on the mixed-mode data collection process used by TurkStat, including computer-assisted telephone interviewing (CATI) and computer-assisted personal interviewing (CAPI), and we provide details on the sampling design (stratified cluster sampling, PPS selection, and weighting procedures).

Comment 6:

4. Discussion

(1) The depth of the argument is insufficient. It only stays at the level of correlation without exploring the causal relationship and fails to put forward a strong and well-documented argument to explain why these relationships exist. For example, gender: Is it because men are more adventurous? Or is it because the platform's algorithm itself has gender bias? Or are women holding back due to safety concerns? More specific mechanisms are needed rather than general discussions about "social structure". For instance, age: Is it because of low digital literacy, lack of demand, or because the interface design is not friendly to the elderly that the elderly keep away from digital platforms? Therefore, it is suggested that they be distinguished and discussed in depth.

Response 6: Thank you for the comments. We fully agree that the Discussion section should provide deeper theoretical and behavioral explanations for the observed relationships. Accordingly, we substantially expanded and revised the Discussion to explore the underlying mechanisms behind each key sociodemographic factor.

Comment 7:

(2) This part lacks theoretical significance, practical significance, research limitations and future research directions.

Response 7: Thank you for the comments. We expanded the Discussion section by adding explicit paragraphs addressing (1) the theoretical significance of the study, (2) the practical implications for policymakers and platform operators, (3) the limitations of the research, and (4) future research directions. These additions enhance the clarity, completeness, and scholarly contribution of the Discussion (see the last four paragraphs of the Discussion section).

Comment 8:

Finally, the article lacks a theoretical basis and research hypothesis section, has an unreasonable structure arrangement, and weak theoretical support. It is suggested to supplement relevant theories to support the article. I suggest the author to improve their manuscript by reviewing and adding recent literature, and finally, I would like to encourage the author to improve their manuscript. I believe that the above points need improvement I once again congratulate the authors of this study and hope that my comments are constructive and helpful for the development of your manuscript.

Best regards.

Response 8: Thank you for the comment. These comments have been very helpful in reviewing and improving our manuscript. We have carefully revising these instructive comments and made corrections that we hope will be approved. The revised parts are highlighted in red in the main paper.

Reviewer #2:

Thank you for the opportunity to review this manuscript. I outline minor suggestions below:

Comment 1: There are few studies examining consumer-level factors in digital platform services (transportation, accommodation, home services) in the Turkish context. In this respect, this study contributes to the literature and has high local specificity.

Response 1: Thank you for the comments.

Comment 2: The findings are valuable for understanding the usage dynamics of digital platform services in Turkey, generating directly applicable insights for policymakers and industry (e.g., the impact of factors such as age, income, and education.

Response 2: Thank you for the comments.

Comment 3: The application of binary logistic regression is well-documented, yet the manuscript lacks comprehensive detail on the sampling methodology.

Response 3: Thank you for the comment. Taking this criticism into account, we have revised the Method. Taking this criticism into account, we have added the necessary explanations.

Comment 4: After reviewing the paper, I find that it adeptly tackles the pertinent issues at hand. However, as a scientific document, it is imperative for the methodology to be clearly delineated. Despite providing a descriptive account of the method, I strongly suggest incorporating an exposition of the applied logistic regression, including its mathematical formulation. This enhancement will enhance clarity significantly. Once this aspect is addressed, the paper will notably distinguish itself.

Response 4: Thank you for the comment. Taking this criticism into account, we have revised the Method. Taking this criticism into account, we have added the necessary explanations.

Reviewer #3:

Comments: This study is a timely and important study that examines the socio-demographic factors influencing service purchasing behavior through digital platforms, specifically in Turkey. The use of the 2023 "Household Information Technology Usage Survey Micro Dataset" provides a strong empirical basis for the study. Furthermore, the binary logistic regression analysis used is an appropriate method for revealing the relationship between the identified variables and platform usage.

Overall, the study offers significant contributions to the literature with its topic, method, dataset, and the timeliness of its findings. It is a valuable resource for both the academic community and public policymakers in understanding and guiding Türkiye's digitalization process.

Response: Thank you for the comment.

My recommendations for the study are below.

Comment 1: It is recommended that you briefly mention the binary logistics method in the Econometrics Method section.

Response 1: Thank you for the comment. Taking this criticism into account, we have revised the Method. Taking this criticism into account, we have added the necessary explanations.

Comment 2: In the conclusion section, the limitations of the study can be mentioned in the last paragraph.

Response 2: Thank you for the comments. Taking this criticism into account, a dedicated limitations of the study has been added.

---

## [Decision Letter · Decision Letter 1]

10 Mar 2026

Factors associated with Turkish individuals’ purchase of transportation, accommodation and household services from digital platforms

PONE-D-25-20784R1

Dear Dr. Alkan,

We’re pleased to inform you that your manuscript has been judged scientifically suitable for publication and will be formally accepted for publication once it meets all outstanding technical requirements.

Kind regards,

Tachia Chin

Academic Editor

PLOS One

Additional Editor Comments (optional):

Reviewers' comments:

Reviewer's Responses to Questions

**Comments to the Author**

Reviewer #2: All comments have been addressed

Reviewer #4: All comments have been addressed

2. Is the manuscript technically sound, and do the data support the conclusions?

Reviewer #2: Yes

Reviewer #4: Yes

3. Has the statistical analysis been performed appropriately and rigorously?

Reviewer #2: Yes

Reviewer #4: Yes

4. Have the authors made all data underlying the findings in their manuscript fully available?

Reviewer #2: Yes

Reviewer #4: No

5. Is the manuscript presented in an intelligible fashion and written in standard English?

Reviewer #2: Yes

Reviewer #4: Yes

Reviewer #2: The study addresses a relevant and timely research question and makes a meaningful contribution to the literature. The research objectives are clearly defined, the methodology is appropriate and well justified, and the data analysis is conducted rigorously. The results are presented in a clear and coherent manner, and the conclusions are well supported by the findings.

The manuscript is well structured, the language is clear and academic, and the references are adequate and up to date. I find that the authors have sufficiently addressed all essential methodological and conceptual requirements expected for publication in this journal.

Based on my evaluation, I believe that the manuscript meets the scientific and academic standards of the journal and is suitable for publication in its current form.

Therefore, I recommend acceptance for publication.

Reviewer #4: The authors included all the comments in the new draft. The authors answered each of the comments. No additional comments.

**Do you want your identity to be public for this peer review?** For information about this choice, including consent withdrawal, please see our Privacy Policy

Reviewer #2: No

Reviewer #4: No

---

## [Editor Report · Acceptance letter]

PONE-D-25-20784R1

PLOS One

Dear Dr. Alkan,

I'm pleased to inform you that your manuscript has been deemed suitable for publication in PLOS One. Congratulations! Your manuscript is now being handed over to our production team.

Kind regards,

on behalf of

Dr. Tachia Chin

Academic Editor

PLOS One